# Mechanisms and Pharmaceutical Action of Lipid Nanoformulation of Natural Bioactive Compounds as Efficient Delivery Systems in the Therapy of Osteoarthritis

**DOI:** 10.3390/pharmaceutics13081108

**Published:** 2021-07-21

**Authors:** Oana Craciunescu, Madalina Icriverzi, Paula Ecaterina Florian, Anca Roseanu, Mihaela Trif

**Affiliations:** 1National Institute of R&D for Biological Sciences, 296 Splaiul Independentei, 060031 Bucharest, Romania; oana.craciunescu@incdsb.ro; 2The Institute of Biochemistry of the Romanian Academy, 296 Splaiul Independentei, 060031 Bucharest, Romania; madalina.icriverzi@biochim.ro (M.I.); florian@biochim.ro (P.E.F.); anca.roseanu@biochim.ro (A.R.)

**Keywords:** liposomes, osteoarthritis, intra-articular, polysaccharides, polyphenols, anti-inflammatory activity

## Abstract

Osteoarthritis (OA) is a degenerative joint disease. An objective of the nanomedicine and drug delivery systems field is to design suitable pharmaceutical nanocarriers with controllable properties for drug delivery and site-specific targeting, in order to achieve greater efficacy and minimal toxicity, compared to the conventional drugs. The aim of this review is to present recent data on natural bioactive compounds with anti-inflammatory properties and efficacy in the treatment of OA, their formulation in lipid nanostructured carriers, mainly liposomes, as controlled release systems and the possibility to be intra-articularly (IA) administered. The literature regarding glycosaminoglycans, proteins, polyphenols and their ability to modify the cell response and mechanisms of action in different models of inflammation are reviewed. The advantages and limits of using lipid nanoformulations as drug delivery systems in OA treatment and the suitable route of administration are also discussed. Liposomes containing glycosaminoglycans presented good biocompatibility, lack of immune system activation, targeted delivery of bioactive compounds to the site of action, protection and efficiency of the encapsulated material, and prolonged duration of action, being highly recommended as controlled delivery systems in OA therapy through IA administration. Lipid nanoformulations of polyphenols were tested both in vivo and in vitro models that mimic OA conditions after IA or other routes of administration, recommending their clinical application.

## 1. Introduction

Osteoarthritis (OA) is a complex, multifactorial degenerative disease of the joint, characterized by chronic inflammation, progressive loss of articular cartilage, subchondral bone sclerosis and osteophyte formation, changes in the synovial membrane and increased volume of synovial fluid with altered coefficient of friction [1,2,3,4,5,6]. In some respects, it can also be viewed as an inflammatory disease, leading to chronic pain and decrease of life quality [5,7,8]. During OA progression, the degradation process of the collagen network takes place constantly and a variety of inflammatory mediators are detected in the articular cartilage. Tumor necrosis factor alpha (TNF-α) and interleukin-1β (IL-1β) influence chondrocyte metabolism, and also induce the production of inflammatory mediators, such as nitric oxide (NO), and prostaglandin E2 [9]. In these conditions, cartilage is further degraded and the inflammatory process is perpetuated [1,10,11]. Several studies indicated that local joint inflammation (synovitis) induced by endogenous molecular products derived from cellular stress and extracellular matrix degradation acted through innate inflammatory network and could influence the integrity and function of articular cartilage [12,13]. On the other hand, the systemic inflammation resulting from metabolic disturbance could also contribute to OA progression [13,14]. Some reports presented OA as a systemic disease and described the complexity of the involved inflammatory mechanisms [15]. Currently, there are no efficient treatments that can stop the pathological processes involved in OA progression, but prevention strategies and treatments directed to symptoms, pain relieve and function regain [16,17]. Treatments are based on various pharmacologic agents, such as selective cyclooxygenase-2 (COX-2) inhibitors, non-steroidal anti-inflammatory drugs (NSAIDs), corticosteroids, even analgesics [18]. Their administration through oral route involves limited bioavailability and risk of side effects, such as upper gastrointestinal and cardiovascular complications [19]. As OA has a localized nature, intra-articular (IA) administration of drugs provides the opportunity to improve the treatment by local depot formation and prolonged drug action [13,20,21,22,23,24]. Although numerous disease-modifying OA drugs (DMOADs) showed promising results in preclinical trials, their poor IA bioavailability limited the treatment approval [25]. In the last years, natural bioactive molecules (e.g., glycosaminoglycans (GAGs) from animal sources or plant polyphenols) have gained considerable interest as therapeutic alternatives [19,26,27,28,29,30,31,32,33,34,35] (see Section 2 for detailed anti-inflammatory activity and Section 5 for mechanisms of action). However, the efficacy of different anti-inflammatory bioactive molecules administration is limited due to their poor stability in the harmful biological milieu or low solubility, which decreases their bioavailability. Several delivery systems, including liposomes, microparticles, nanoparticles and hydrogels have been investigated for the sustained delivery and controlled release of bioactive molecules in the joints [5,36,37] (see Section 4 for lipid nanoformulation and Section 5 for mechanisms of action). In vitro studies have demonstrated that in the case of OA, the phospholipidic layer acting as a boundary lubricant was missing from the articular surface of osteoarthritic degenerated cartilage and the structure of chondroitin sulfate (CS) was also changed [6]. Liposomes are the most commonly used nanocarriers to deliver drugs to human tissues in clinical applications and have been approved by the US Food and Drug Administration (FDA) [38] (see Section 3 for preparation and advantages)**.** As drug carrier systems, liposomes possess many biophysical and physicochemical properties suitable for IA administration, such as sustained release, ability for self-assembly and capacity to load large quantities of drugs [23,37]. Additionally, due to their ability to incorporate hydrophilic and hydrophobic molecules, good biocompatibility, low toxicity, activation and targeted delivery of bioactive compounds to the site of action, liposomes offer many advantages, such as the protection and efficiency of encapsulated material, solubilization of lipophilic molecules, prolongation of the duration of action and present targeting options. The clinical development of liposome-based drug delivery systems with synergistic therapeutic effect and a description of the technologies for NSAIDs liposomal formulations for orthopedic field applications were previously reviewed [39,40,41,42]. The only product approved in Germany and available on the market for IA administration in human patients with OA is Lipotalon^®^, containing the liposomal formulation of dexamethasone-21-palmitate [43].

In this review, we have focused on natural bioactive compounds of animal and plant origin that have been proposed or examined to date for the treatment of OA. Data regarding their anti-inflammatory activity and mechanisms of action, and their lipid nanoformulation as efficient delivery systems are the main topics reviewed here. The novel trends in liposomes preparation and advantages for IA therapy of OA are also discussed.

## 2. Natural Bioactive Compounds with Anti-Inflammatory Activity and Mechanisms of Action

### 2.1. Polysaccharides with Anti-Inflammatory Activity

Glucosamine (GLU) is an amino sugar, present as a component of chitosan and chitin in crustacean shells, which has been marketed for decades as a nutraceutical for supporting the structure and function of OA patients’ joints. GLU was included in the group of symptomatic slow-acting drugs of OA (SYSADOA) and presented long-term beneficial effects (6–24 months), decreasing the Western Ontario and McMaster Universities Osteoarthritis index (WOMAC) scores for pain and stiffness, in particular in combination with chondroitin sulphate (CS) [44]. Good safety was reported as an advantage over fast-acting analgesics, such as NSAIDs and corticosteroids, which presented low gastrointestinal tolerance [45] at a comparable pain relief level.

GAGs are a family of long chain polysaccharides with a linear structure, consisting of repetitive hexosamine-uronic acid units highly sulphated in variable positions, excepting hyaluronic acid or hyaluronan (HA). They are the basic constituents of proteoglycans from the extracellular matrix of cartilage and fulfil different biological tissue-specific roles, according to their various structures. Thus, CS is a sulphated constituent representing 80% of the GAGs in adult cartilage, but aging and disease modify its structure and function [46]. The insignificant immunogenicity of pharmaceutical-grade products [47] together with their anti-inflammatory, antioxidative and anti-thrombosis activity were explained on a biochemical basis in several in vitro and in vivo models, recommending the use as a supplement for maintaining the integrity of OA articular tissue [48].

The described CS interaction with cytokines, chemokines and growth factors involved in the progression of inflammation was based on its highly negative charge [49]. The pharmacodynamics of highly purified CS in human articular chondrocytes consisted of the decrease of nuclear factor-kB (NF-kB) translocation through p38 mitogen-activated protein kinase (p38 MAPK) and extracellular-signal-regulated kinase 1/2 (Erk1/2) signalling pathways [33]. Thus, CS could inhibit the secretion of pro-inflammatory cytokines IL-1 β and TNF-α, pro-inflammatory enzymes phospholipase A2, cyclooxygenase 2 and nitric oxide synthase-2 [33], along with lowering the synthesis of matrix metalloproteinases (MMPs) and their activity [50]. The suppression of inflammatory cytokines in macrophage-like cells was found to be exerted by CS via toll-like receptor-4 (TLR4), regardless of its size, but the GAG–receptor relationship is still under research [49].

Consequently, CS showed anti-inflammatory and immunomodulatory effects in OA patients, its efficacy and safety indicating an economic impact over the use of NSAIDs [33]. Oral administration of CS, alone or in combination with GLU and methylsulfonylmethane (MSM) as nutraceuticals in OA patients with moderate to severe pain showed a bioavailability of 5–15% CS and 26–44% GLU, respectively, its pharmacokinetics being influenced by the molecular weight, sulfation degree and charge density [44]. CS plays an important role in confining water content and providing resistance and elasticity in the OA-degraded articular cartilage [51]. Significant evidence and advantages of the role of CS in ameliorating the pathogenesis of knee cartilage, synovium and subchondral bone in OA patients are reviewed in Table 1. However, its symptomatic efficiency in OA patients is still controversial, mainly in relation to the product brand and patients’ condition, requiring novel strategies of administration.

HA is the major GAG secreted by cartilage cells in the articular fluid, presenting both a high molecular weight and a unique structure, differentiated by the lack of sulfation and core protein binding. Owing to these structural characteristics, HA exhibits rheological properties of great importance for lubrication, viscoelasticity, shock absorption and nutrition of joints. HA is approved by the U.S. FDA and European Medicines Agency as a safe viscous supplement used in OA treatment, mainly through IA administration in patients not responding to NSAIDs or analgesics, such as acetaminophen [20]. Several studies indicated multiple pharmacologic actions in OA models through antioxidative, anti-inflammatory, analgesic and cartilage repair activities [68]. Its anti-arthritic effects are based on the interaction of HA with cell surface receptors, such as CD44, hyaluronate-mediated motility and intercellular adhesion molecule 1 (ICAM-1), which, together with HA binding to layilin, a talin-binding hyaluronan receptor identified in human chondrocytes, contribute to OA condition improvement [69]. The beneficial chondroprotective effect of HA and its clinical efficiency in maintaining cartilage’s integrity are reviewed in Table 1. Although pharmacokinetic data are sparse, it is emphasized that the IA route avoids the gastrointestinal barrier encountered in oral HA administration, offering increased bioavailability [70,71]. Moreover, joint retention for 12–24 h following IA delivery of HA could augment the lubricating properties of the synovial fluid. There are still controversies regarding HA efficiency in relation to its molecular weight, viscoelastic properties, and injections frequency due to rapid plasma elimination, which diminishes its therapeutic role. The heterogeneous response across population requires novel solutions based on nanostructured drug delivery platforms, such as liposomes, mainly used for IA delivery of HA [24,72].

### 2.2. Plant Polyphenols with Anti-Inflammatory Activity

Polyphenols are natural phytochemical compounds produced in plants as secondary metabolites. As a consequence of their structure based on one or more aromatic rings and hydroxyl groups, they possess several bioactivities, such as antioxidant, anti-inflammatory, antimicrobial, antiaging, antitumor and cardioprotective activity [73]. Traditional medicine uses polyphenols as nutrients or locally administered products for the treatment of arthritis. Although an increased number of studies have demonstrated the role of polyphenols as antioxidant and anti-inflammatory agents, their mechanism of action at the chondrocytes level remains unclear. These studies did not show any effects of polyphenols on the intensity of joint inflammation, but confirmed their binding to collagen molecules from the articular cartilage, creating stable crosslinks through hydrogen bonding and increasing the resistance of constituents to enzymatic attack [74]. In addition, polyphenols were able to inhibit the inflammatory cytokines, blocking the cascade of reactions that could lead to cartilage damage, thus exerting a chondroprotective effect [75].

The cartilage protective effect of several polyphenolic compounds, such as flavonoids, catechins, tannins, phytoalexins (stilbenoids) and curcuminoids were investigated by both in vitro incubation with chondrocytes or cartilage explants and in vivo IA injections in CIA models, as detailed in Table 1. The bioavailability of polyphenols after oral consumption was mainly due to absorption in the gastrointestinal tract and the reported increase of plasma antioxidant activity, ensuring the prevention of chronic disease [76]. In turn, IA injections of polyphenols in an arthritic rat model showed a more beneficial effect in the treatment of acute OA [35].

Among polyphenolic compounds, flavonoids were highly potent in the modulation of cartilage degeneration [77]. Several sub-classes of flavonoids, such as (i) flavanols or catechins, in particular epigallocatechin-3-o-gallate (EGCG); (ii) flavones, such as apigenin and luteolin; (iii) flavonols, such as quercetin, kaempferol, morin and their glycoside derivatives; and (iv) phytoalexins, such as resveratrol are widely distributed in green tea, wine, nuts, berries and dark chocolate, revealing the potential to switch the phenotype of macrophages [78]. In addition, flavonoids found in certain plant species, such as curcumin, silibinin, chrysin, baicalin, wogonin, hyperin, and alpinetin suppressed the expression of several metalloproteinases (MMPs) involved in cartilage degradation [63,64]. The anti-inflammatory activity of catechins induced the decrease of protein denaturation and cytokine production, while their antioxidant activity was shown as ROS damage and stimulation of the synthesis of antioxidant enzymes, thus indicating an important role in arthritic inflammation reduction [79]. Tannic acid injections reduced paw edema in rats by decreasing myeloperoxidase enzyme activity, similar to indomethacin treatment [80]. Resveratrol was an efficient adjuvant in meloxicam treatment of patients with knee OA, reducing serum inflammatory cytokines, C-reactive protein and complement proteins C3 and C4 in a significant manner, compared to a placebo-treated group and decreasing the pain [81]. Alkaloids, as with oxymatrine, prevented inflammation by suppressing NF-kB, p38 MAPK and TLR4 signalling pathways [82]. Not least, a synergistic effect could be obtained through the combination of polyphenols combinations to increase their anti-inflammatory activity [30,67].

Spices, such as ginger (Ziangiberaceae), which contains gingerol, improved the mobility and pain reduction in clinical trials [17,83]. Highly purified extracts of *Uncaria tomentosa* (cat’s claw) (Rubiaceae) and *Harpagophytum procumbens* (devil’s claw) (Pedaliaceae) containing harpagoside as an anti-inflammatory component showed variable efficiency in the remission of knee OA stiffness in clinical tests on OA patients after 4–12 weeks administration [84]. These studies reported the inhibition of TNF-α, IL-6 and IL-1β cytokines, decrease of PGE2 production, interference with eicosanoid biosynthesis and, finally, COX-2 inhibition and inflammation control. In addition, *Tamus communis* (Dioscoreaceae), *Arnica montana* (Asteraceae) and *Symphytum officinalis* (Boraginaceae) could suppress the expression of MMPs and COX-2, mainly through the blocking of the NF-kB pathway [85].

## 3. Recent Advances in Liposomes Technology

Lipid nanostructures are the most investigated carriers for targeted drug delivery [39,86,87,88]. Among them, liposomes are artificially prepared vesicles made of phospholipid bilayers [86,89] having the potential for the treatment of arthritic diseases [90,91,92,93]. The liposomal encapsulation of natural compounds or drugs was proposed by several researchers and many data in the field demonstrated the improvement of their performances [6,94]. Higher retention and slower release of bioactive agents from liposomal formulations were found in direct relation to the type of liposomes, lipid composition, size and charge, and lipid–drug ratio, indicating the need for novel systems and the development of novel strategies. Liposomal formulations used in clinical trials or already on the market were previously described by Bulbake et al. [88]. In this paper, recently researched liposome manufacturing technologies and liposomal formulation advantages are reviewed.

### 3.1. Novel Methodological Approaches

Several techniques have been reported for the preparation of liposomes with high entrapment efficiency, narrow particle size distribution and long-term stability, i.e., the Bangham method, detergent depletion, ether/ethanol injection methods, reverse-phase evaporation and the emulsion method [37,87,95,96,97,98]. Liposomes prepared using the thin layer hydration method [99] had a rather high size (10–100 µm), required expensive post-processing steps to reduce the vesicles diameter and frequently presented physical degradation, drug leakage or aggregation. However, it remains the most commonly used method for the preparation of liposomes. The emulsion method had as main drawback the large amount of organic solvent remaining in the suspension [100], while reverse phase evaporation was limited by the long-lasting drug–solvent contact, leading to drug denaturation [101].

A method proposed for increasing the liposome stability and entrapment efficiency was based on the freeze–thaw procedure. Application of a series of freeze–thaw cycles could break the multilamellar vesicles into unilamellar vesicles, in order to obtain the targeted size and to maximize the encapsulation efficiency [23,102]. A process that could inhibit the aggregation of vesicles and the degradation of phospholipids during storage was based on the freeze-drying or spray-drying of a cryoprotectant–liposomes mixture to obtain a dried liposomal formulation. The method was lately optimized to allow high encapsulation efficiency of polyphenols, such as ECGC, quercetin, in phosphatidylcholine/cholesterol liposomal formulations for prolonging their stability during storage, an important property for pharmaceutical use [103,104]. This dried formulation ensured the reconstitution of lipid vesicles after reconstitution in solution, but some modification of their physicochemical properties took place and the quality control was recommended for its translation to the industrial level [105].

A novel supercritical CO_2_-assisted process of the preparation of liposomes consists of water droplets generation followed by the formation of a lipid layer around them with minimal solvent residue [106]. Formation of phosphatidylcholine/cholesterol and phosphatidylcholine/phosphatidylethanolamine liposomes allowed 98% encapsulation efficiency of an antioxidant purine base able to enhance the anti-inflammatory activity of steroids and 6 months stability [107]. In addition, the drug release rate was improved through the entrapment of cholesterol and phosphatidylethanolamine next to phosphatidylcholine in the lipid bilayer of the vesicles, which thus became more compact [108]. To control the nanometric diameter of vesicles, the ratio between gas:liquid flow rates was also optimized, reporting high liposome size ranging between 870 and 1730 nm for low ratio values (<1.8) and small vesicles of 126–139 nm for high values (>6) [109]. Both hydrophilic and lipophilic compounds could be loaded using the gas-based process with good efficiencies (~75%), indicating the possibility of pharmaceutical applications. Thus, unilamellar stable liposomes encapsulating a low soluble drug were prepared by the supercritical fluid CO_2_ method from phospholipids and cholesterol, showing small diameters (137 nm), high encapsulation efficiency (90%) and pharmacokinetics equivalent to the commercial formulation [110]. The authors reported that the addition of ascorbic acid to the liposomal formulation ensured minimal changes in size and zeta potential after freeze-drying under vacuum, at −88 °C, for 24 h and rehydration with normal saline.

A one-step procedure based on microfluidics technology represents an alternative strategy for liposome manufacturing. Several microfluidic devices with chip-based or capillary-based structures ensured high-throughput preparation of liposomes in a well-controlled and reproducible manner in terms of size distribution [111]. Higher entrapment efficiency of hydrophilic and lipophilic drug in liposomes was reported in the case of using microfluidics, compared to traditional methods, but thorough selection of input parameters, such as solvent, flow rate, temperature or chip design had to be considered [112]. Novel microfluidics protocols that prevent vesicles aggregation and the instability of liposomes were recently established [113]. Moreover, the role of lipid composition was systematically investigated as an important factor for liposome formation by microfluidic mixing [113]. Additional advantages of microfluidics technology over traditional laboratory methods, such as low sample volumes, reduced costs, no disruption stage after liposomes manufacturing, were recently assessed [114]. This technology has a high potential to produce nanoscale lipid vesicles for medical applications, but several factors have been reported as limiting its translation from the lab to the industrial scale [114].

### 3.2. Advantages and Limits of Using Liposomes as Efficient Delivery Systems

To date, liposomes have been the most successful formulation of drugs for clinical applications and the sterically stabilized liposomal formulations are the most FDA-approved products [40,115,116]. Data reviewed from the literature showed that 15 drugs with liposomal delivery system are currently available on the market and more than 40 additional liposomal drugs are under clinical trials [39,40,88,97,117]. Liposomes offer several advantages over other delivery systems, including biocompatibility, control of biological properties via modification of the physicochemical properties, such as lipid composition (in particular, the presence of cholesterol and stearylamine), vesicle size, lamellarity, positive surface charge and lipid membrane fluidity. All these are important characteristics that modulate the therapeutic role and interaction with cells by several mechanisms of drug delivery [41,87,89,98,118,119,120,121].

A major advantage of using liposomes is the larger quantity of loaded bioactive agent that passes through the cell membrane into the cytoplasm. Moreover, lipophilic compounds, such as polyphenols, could be solubilized by entrapment into the lipid bilayer of the liposomes [122]. In addition, many reviews highlighted the important role of liposomes as such in boundary lubrication of joints and protection of articular cartilage from degenerative changes [123,124,125,126].

Stability of the liposomal formulations in physiological conditions is a key issue in drug delivery [39]. In order to increase the stability of liposomes’ in the presence of the synovial fluid after IA administration, it was necessary to include polar lipids in the bilayer, as has been reported [87,127]. Additionally, cholesterol incorporation in the bilayer enhanced the membrane stability and the encapsulation efficiency of both hydrophilic and lipophilic bioactive molecules in the liposomes [128]. Increased cholesterol concentration in the phospholipid bilayer could lead to a gradual disappearance of the phase transition without affecting the transition temperature [129,130].

### 3.3. IA vs. Oral Administration

IA administration of bioactive agents is considered a direct and better route of delivery, being recommended as a more efficient therapy for joint cartilage, compared to oral administration associated with side effects due to systemic bioavailability [131]. The clinical performances and advantages of IA drug applications were previously presented in a series of well-documented studies [20,24,132]. Thus, IA administration ensured local biodistribution of bioactive compounds in cartilage tissue. However, the kinetics of the clearance of compounds from the joint varies, according to their chemical structure [133] and size [134]. Therefore, to improve local bioavailability of compounds, the researchers highlighted their delivery to the deeper zone of the articular cartilage [135]. Additionally, to increase the residence time of compounds, IA administration through repeated injections was recommended. However, some limitations of IA administration have been reported, including adverse joint events that could accelerate OA progression due to infection, osteonecrosis or subchondral fracture [136,137].

Currently, natural bioactive compounds, such as lubricin (proteoglycan 4) [138], hyaluronan [139], EGCG [140], resveratrol [141,142] and curcumin [143,144] are under investigation for their lubrication, anti-inflammatory and antioxidant potential and to establish an adequate route of administration, alone or as adjuvants to potentiate the usually used NSAIDs or analgesics. All these studies showed that oral consumption minimizes the function of compounds due to intense metabolization in liver and gut, resulting in degradation and low bioavailability. Topic administration of lipophilic compounds is prevented by the skin barrier, limiting their bioavailability. On the contrary, IA administration of GAGs and polyphenolic compounds into the synovial space could offer improved bioavailability at the damaged site. In addition, their phamacokinetics, biodistribution and anti-inflammatory properties might be beneficially modified using liposomal formulation technology, which enables easy cellular penetration due to similar phospholipid composition to cell membrane [23] and the significantly extended drug residence time within the joint [23,145].

## 4. Lipid Nanostructures Loaded with Natural Anti-Inflammatory Compounds for OA Treatment

In general, natural bioactive compounds with anti-inflammatory properties lack water solubility and chemical stability, but lipid nanoformulations of their derivatives showed improved therapeutic benefits [146]. In order to be used in OA treatment, GLU, CS, lactoferrin and HA were the main natural compounds of animal origin with anti-inflammatory activity encapsulated in various lipid nanostructures and evaluated for improved bioavailability, activity and barrier penetration. Previous studies have demonstrated the in vitro and in vivo biocompatibility as well as the anti-inflammatory activity of CS alone or its encapsulated formulation in lipid nanostructures [34]. Association of CS with analgesic drugs, such as diacerein or tapentadol loaded in nanovesicles improved local therapy of OA [147]. Emerging strategies for the IA administration of CS or HA targeted their use as homing carriers by the modification of the surface of liposomes, delivery of encapsulated anti-inflammatory drugs into the synovium membrane, articular space or osteochondral lesions and the limitation of the cytotoxicity and adverse effects, compared to free drugs [5].

In view of OA therapy at the articular cartilage level, biolubricants were also necessary and phosphatidylcholine-based liposomes could act as such or as drug delivery systems to increase the uptake of loaded HA or lubricin [148]. A more realistic model was lately reported to highlight the synergistic effect of three components—HA, lubricin and phosphatidylcholine lipid—which conferred efficient boundary lubrication as hydration layers and easy restoration upon shear kinematics [149]. This observation indicates the importance to deliver lipid-based nanostructures encapsulating HA or other compounds of interest, to ensure both lubrication and an anti-inflammatory effect using an IA injectable route of administration for improved and efficient OA therapy [6].

In vivo comparative studies on DBA1 CIA mice using IA administration of lactoferrin, an anti-inflammatory bioactive molecule, revealed that the encapsulation in positive multivesicular liposomes protected it from severe proteolysis and demonstrated the controlled release and better compound retention at the damaged site in comparison to free protein, in [23,150]. Several representative studies on lipid-encapsulated GAGs and lactoferrin are reviewed in Table 2.

In recent years, natural compounds with anti-inflammatory activity isolated from medicinal plants were formulated by encapsulation in lipid nanostructures, highlighting their use as promising systems for OA therapy. Besides liposomes, other lipid nanoformulations, such as solid lipid nanoparticles or nanogels, were designed for polyphenolic compound loading, in order to expand their biomedical applications [93,167]. Several advantages, such as increased stability, biocompatibility and cellular uptake, were revealed in the case of lipid nanoformulations over free compounds [167]. The main purpose of protection and increased bioavailability was fulfilled by catechin encapsulation in nanoparticles, suggesting improved delivery conditions [168]. High encapsulation efficiency was obtained for resveratrol incorporated in extruded liposomes together with slow diffusion and preservation of its antioxidant activity [169]. Important curcuminoid anti-inflammatory compounds presenting poor water solubility were loaded in phosphatidylcholine liposomes to enhance their cellular uptake and provide a protective effect by in vitro inhibition of inflammation and prevention of osteoclastogenesis in LPS-stimulated macrophages [162]. In vitro treatment of chondrocytes with two anti-inflammatory polyphenols—resveratrol and curcumin—co-encapsulated in lipid-core nanocapsules showed reduction of the nitric oxide level and hindrance the cell membrane of inflamed chondrocytes to suffer apoptosis characteristic changes, usually observed during OA progression [30]. In addition, in vivo studies on the treatment with resveratrol and curcumin encapsulated in lipid-core nanocapsules for 8 days by injection in rats with induced arthritis indicated a better antioedematogenic effect than non-encapsulated compounds and lack of hepatotoxic effect [170]. Several representative studies regarding lipid nanoformulations of natural compounds of vegetal origin are reviewed in Table 2.

## 5. Anti-Inflammatory Activity and Mechanisms of Action Demonstrated in Experimental Models In Vitro and In Vivo

Certain anti-inflammatory polysaccharides from animal sources (GLU, CS, HA) and plant bioactive compounds (phenolic acids, flavonoids, phytoalexins, curcumin, tannins, alkaloids) are either orally or IA administered into OA knee, alone or in combination with NSAIDs or analgesics, serving as both prophylactic and therapeutic agents for the protection of the articular cartilage [85]. Novel formulations can be obtained by their loading in lipid nanostructures that could enhance their anti-inflammatory activity and improve OA therapy, due to better solubility and stability [146]. In addition, this strategy provides the controlled release of loaded natural compounds and their protection against the enzymes present in the milieu on the path, changing the pharmacokinetics and bioavailability in vivo [171]. On the other hand, lipid nanostructures administered by oral, topic or IA routes have the capacity to modify the cell response in OA lesions [134]. Three mechanisms of action were described: (i) the decrease of ROS and nitric oxide level; (ii) inhibition of MMPs, disintegrin and metalloproteinase with thrombospondin motifs (ADAMTSs); and (iii) reduction of pro-inflammatory cytokines and inflammasome production. Macrophages remain the main target in OA treatment, including all immunomodulatory cell subpopulations presents in the joint [172]. Several key molecules, such as IL-1 β, IL-6, TNF-α, MMP-3, type II collagen, aggrecan or osteopontin, were identified as targets of the antiarthritic therapy, their mechanisms of action depending on the form of arthritis [147].

### 5.1. Redox Control

Formation of excessive reactive oxygen species, nitric oxide and advance glycation end products leads to oxidative stress- and age-related diseases with important damaging potential at chondrocytes level, playing a role in the development/pathogenesis of OA [27]. Thus, natural antioxidant agents consumed as nutraceutics or locally administered are recommended to interfere with inflammatory reactions, although little evidence is provided on their transport to cartilage. It was reported that HA supplementation in stressed bovine articular chondrocytes was involved in redox control, reducing ROS through the regulation of nuclear factor-erythroid-2-related factor (Nrf2) expression via Akt phosphorylation and considered an important protective mechanism in OA pathogenesis [173]. GLU also exerted chondroprotection via the reduction of nitric oxide production [174]. Free HA and CS served as surface ligands of NSAIDs-loaded solid lipid nanoparticles to ensure the reduction of knee joint inflammation in OA rats by IA administration [156,175].

Plant-derived bioactive molecules, in particular, polyphenols, known as amphipathic compounds, easily bind to the cell membrane of chondrocytes, diffuse and enter into the cytoplasm, enabling antioxidative effect in cytosol or nucleus through scavenging of reactive oxygen species and nitric oxide [176]. EGCG was found to be better than curcumin and quercetin in the modulation of OA processes related to oxidation, inflammation and finally, joint degradation [177]. EGCG and its methylated derivatives decreased nitric oxide levels in pre-treated human OA chondrocytes, along with reducing the expression of multiple markers, such as enzymes, angiogenic factors and cytokines, indicating anti-arthritic effects [26,35]. In vivo primary OA model related to aging in guinea pigs confirmed the beneficial action of IA treatment with EGCG on reducing ECM degradation, inflammation and cell senescence [140]. Lipid nanoparticles and liposomes were tested as EGCG carriers, showing good interaction and encapsulation efficiency, increased the bioavailability and also potent anti-inflammatory activity, which justified further studies despite the cost of nanotechnology [178].

Resveratrol and quercetin co-loaded in lipid vesicles showed therapeutic efficacy in vivo against chemically induced oedema, mainly through antioxidant activity of polyphenolic compounds and the facilitation of cellular uptake through liposomal formulation [159]. Co-encapsulation of resveratrol and curcumin in 200 nm lipid-core nanocapsules proved prolonged antioxidant activity of each compound against reactive oxygen species generated in vitro [179] and the antioedematogenic activity in a rat arthritis model, suggesting a synergistic therapeutic effect [170].

In innate immune cell types, curcumin showed nitric oxide inhibition through NF-kB, JNK and p38 signalling pathways [180]. The liposomal formulations of curcumin maintained the potential to inhibit nitric oxide production in RAW264.7 macrophages through the suppression of cathepsin K and TRAP expression [162]. An argument for lipid nanoformulation was the observation that nitric oxide inhibition potential was also imparted by empty liposomes due to their phosphatidylcholine content. A synergistically anti-inflammatory effect of the natural compounds mixture of curcumin, EGCG and hydrolysed collagen in IL-1β-stimulated chondrocytes, inhibiting the oxidative stress generation by modulation of the antioxidant enzymes system and catabolic mediators, revealing potential for OA patients therapy, was reported [181].

### 5.2. Modulation of Catabolic Mediators

Several MMPs, as with MMP-1, MMP-2, MMP-3, MMP-7, MMP-8, MMP-9 and MMP-13, are the main enzymes involved in cartilage degradation [182]. CS and HA were shown to inhibit MMP-3 synthesis in human osteoarthritic chondrocytes [50]. In vitro studies using chondrocyte lines and cartilage explants demonstrated the capacity of flavonoids to suppress MMPs expression through mechanisms not involving mitogen-activated protein kinase (MAPK) inhibition, signalling molecules, such as nuclear transcription factor-κB (NF-κB), nuclear factor erythroid-derived 2-related factor 2 (Nrf2) and phosphatidylinositol 3 kinase/protein kinase B (PI3K/Akt) [64]. The chondroprotective effect of flavonoids by blocking ADAMTSs expression, key enzymes that degrade aggrecan during OA was reported on mice OA model [62]. The IA injection of resveratrol suppressed the expression of MMP-13 and iNOS catabolic factors by SIRT1 activation in mouse OA cartilage [141]. The same treatment proved the chondroprotective effect of resveratrol by chondrocyte autophagy and cartilage degradation delay through the modulation of HIF-1α and HIF-2α expression balance and regulation of AMPK signalling pathway [142]. Curcumin limited MMP-3 secretion in cartilage explants in a dose-dependent manner, mitigating OA progression [183]. Liposomes encapsulating curcumin retained their anti-catabolic activity and downregulated the expression of MMP-3 in IL-1β-stimulated 7F2 osteoblasts, showing their potential to treat OA changes in subchondral bone [162]. In vivo studies in CIA rats showed that treatment with liposomes loaded with dimethyl curcumin inhibited MMP-2 and -9 activity in blood, adjusting the extracellular matrix degradation [166].

### 5.3. Modulation of Cytokines Level

The anti-inflammatory cellular mechanism of natural compounds consists of the modulation of the expression and secretion of cytokines molecules, which play a critical role in the pathological process of OA development [147]. Pro-inflammatory cytokines, such as TNF-α, IL-6, and IL-8, are important mediators of inflammation. Their down-regulation at both gene-level expression and protein production represented the main purpose of therapeutic strategies to prevent OA cartilage destruction. Thus, in vitro studies showed that CS loaded in positive liposomes could inhibit pro-inflammatory cytokines in inflamed human THP-1-derived macrophages and IL-1β-stimulated rabbit chondrocytes [34]. The results conclusively demonstrated a significant increase of the anti-inflammatory capacity, compared to the free compound. Similar observations were reported in case of liposomal lactoferrin formulations suppressing TNF-α production in peripheral blood mononuclear leukocytes [184]. Lactoferrin loaded in positively charged liposomes decreased the pro-inflammatory cytokines produced by lymph node T cells in DBA1 mice with CIA, while increasing anti-inflammatory cytokines IL-5 and IL-10 [23].

Flavonoids, such as wogonin, baicalin and kaempferol, regulated the transcription of IL-1 and IL-6 cytokine genes by NF-kB signalling and interrupted their production, as reported in both in vitro models of inflamed macrophages and in mice with CIA [77]. Although lower potency was observed compared to NSAIDs, flavonoids were desirable in chronic conditions due to their good tolerance in long-term administration. Curcumin was found to inhibit in vitro production of TNF-α in chondrocytes and the degradative processes in cartilage explants [27]. The mechanism of inflammation induction in cartilage degradation during OA and the anti-inflammatory activity of curcumin occurred through the NF-kB pathway [28]. The authors reported that the first stimulated interleukins—IL-1 β and TNF-α—promoted the cascade of cytokines production (IL-1 β, IL-6, IL-8, IL-10) after activation of I kappa beta kinase (IKK), followed by phosphorylation of IKB-α, ubiquitination and stimulation of more than 400 genes in the nucleus. An extended administration of curcumin-phosphatidylcholine complex for 8 months decreasing pain and improving joint function through anti-inflammatory activity proven at IL-1 β and IL-6 level was reported in clinical studies [28].

### 5.4. Modulation of Inflammasome and Toll-Like Receptors (TLR)

The inflammasome is a complex of receptors from the innate immune system cells, such as macrophages, triggered by pathogen-associated molecular patterns (PAMPs) and driven by TLR signalling, activating an inflammatory response [185]. In OA, the reactive species of oxygen present in synovia were suggested as agents that activated NLRP3 inflammasome and IL-1 β, IL-18 cytokines, which amplified the inflammatory response [186]. CS tested in THP-1 differentiated to macrophages attenuated the inflammatory response via inhibition of NF-kB transcription factor, as the main mechanism of action, but did not intracellularly modify IL-1 β/pro-IL-1 β ratio, working upstream of the inflammasome [187]. HA could also play a crucial biological role in innate immunity and attenuated inflammation and tissue damage in a TLR4-dependent manner, although direct binding of TLR to HA was lacking [188].

The inhibitory effects of flavonoids on NLRP3 inflammasome was in direct correlation to TLR4 activation and NF-kB pathway [189]. The expression of NLRP3 components IL-1 β, IL-18, caspase-1 was downregulated by several flavonoids, such as luteolin, hyperin, anthocyanin [31]. In addition, quercetin, EGCG and morin could block the inflammatory mechanisms of arthritis in animal models. The intervention of astilbin in the TLR4/NF-kB signalling cascade in human OA chondrocytes and its chondroprotective effect in mouse OA model has been reported [190]. In vivo studies using an OA model in rats showed that locally administered curcumin by IA injections could suppress the synovial inflammation by down-regulation of TLR4 and NF-κB, suggesting the important role of TLR4 receptors [191]. Based on this observation and corroborated with the data previously reported [162,166], the entrapment of natural compounds into liposomes and IA administration could enhance their inhibitory effect on inflammasome activity, thus impairing OA progression.

### 5.5. Control of Bone Remodelling Processes

Successful results of anti-inflammatory compounds loaded in lipid nanostructures were obtained by modulation of the osteoprotegerin/receptor activator of nuclear factor kB/ ligand (OPG/RANK/RANKL) signalling pathway regulating the osteoblast–osteoclast interaction during bone re-modelling [192]. RANKL is a member of the TNF family and the high OPG/RANKL ratio prevents osteoclastogenesis and stimulates osteoblast differentiation, reflecting a good environment in OA patients. CS treatment in IL-1β-stimulated mice osteoblasts for 7 days counteracted the production of MMP-3, MMP-13 and RANKL, indicating potential not only in cartilage, but also in subchondral bone protection [193]. The inhibition of osteoclast activation in 7F2 osteoblastic cells was proved in the case of treatment with curcumin-loaded liposomes by an increase in the OPG/RANKL ratio and the prevention of RANK binding [162]. Moreover, curcumin and the methylated derivatives, empty liposomes based on phosphatidylcholine and the liposomal formulation of curcumin were reported to suppress MMP-3 and COX-2 expression in human osteoblasts, as potent bone turnover modulators in OA progression.

## 6. Conclusions

Over recent years, delivery systems of biologically active compounds have become more advanced and complex when designing their lipid nanoformulation. Current research activities point towards finding an optimal formulation with suitable properties, capable of delivering encapsulated compounds. Liposomes have shown many advantages as carriers, including increased stability, reduced degradation, enhanced solubility of the drug, and improved pharmacokinetics. Several types of lipid nanocarriers have been developed for the treatment of different diseases, including OA, with increased attention being given to improve the delivery, efficacy, and safety. As we described in this review, lipid nanoformulations, mainly liposomes, loaded with natural bioactive molecules with anti-inflammatory activity, such as polysaccharides, polyphenols and proteins showed increased solubility and bioavailability, being able to improve their therapeutic effect in vivo. These are important aspects for further clinical research and the application of lipid nanoformulations for the treatment of OA.

## Figures and Tables

**Table 1 pharmaceutics-13-01108-t001:** Natural compounds with anti-inflammatory activity used for OA therapy.

Natural Compound	Main Class	Composition/Subclass	Administration	Activity [Reference]
*Polysaccharides*				
GLU	Polysaccharides	Amino sugar	Oral administration of 1500 mg/day in symptomatic OA	Pain relief and improvement of Lequesne index, but not WOMAC score [52]
CS	Polysaccharides	Repetitive sulfated amino sugar and glucuronic acid units	Oral administration of 800 mg/day, 6 months	Better than paracetamol to reduce synovitis [53]
CS	Polysaccharides	Repetitive sulfated amino sugar and glucuronic acid units	Oral administration of 1200 mg or 3 × 400 mg/day, 24 months	Improved joint swelling and WOMAC pain score; better than celecoxib in preserving cartilage volume [19]
CS and GLU	Polysaccharides	-	Oral administration of 400 mg CS and 500 mg GLU, 6 months in OA patients with moderate-severe pain	Reduced pain similar to celecoxib [29]
CS and GLU and MSM	Polysaccharides and sulfur compounds	-	Clinical trial in grade I-II knee OA	Reduced pain and improved WOMAC scores [54]
HA	Polysaccharides	Repetitive amino sugar and glucuronic acid units	Human IL-1 β treated chondrocytes from total knee replacement patients cultured with 1 mg/mL HA and 17 µg/mL P15-1 peptide	Inhibited the catabolic events via MAPK, CD44 clustering and TLR4 signaling, enhanced the protective environment for chondrocytes and stem cells [55]
HA	Polysaccharides	Repetitive amino sugar and glucuronic acid units	Oral HA, 225 mg/day for 2 weeks and 150 mg/day for 2 weeks in patients with knee OA	Relieved knee pain in synovitis [56]
HA	Polysaccharides	Repetitive amino sugar and glucuronic acid	Intra-articular HA for 6 months in patients with knee OA grade II-III	Improved OA symptoms and had a carry-over effect for 1 year [57]
*Polyphenols*				
EGCG	Flavonoids	Catechins	In vitro model of acute injury in H_2_O_2_-treated bovine chondrocytes	Reduced ROS and NO production [35]
EGCG	Flavonoids	Catechins	In vivo IA injection in CIA rats	Articular cartilage resistance to degradation in therapeutic groups [58]
Tannic acid	Phenolics	Tannins	In vivo IA injection in CIA rats	Reduction of cartilage degradation in prophylactic groups and therapeutic groups [58]
Quercetin	Flavonoids	Flavonols	In vitro incubation of bovine articular cartilage explants	Inhibited matrix-degrading enzymes and aggrecan loss [59]
Resveratrol	Phenolics	Phytoalexins (Stilbenoids)	In vitro primary human articular IL-1β treated chondrocytes with	Inhibited cell apoptosis and mitochondria degradation, blocked caspase pathway and reversed ROS up-regulated production [60]
Silibinin	Flavonoids	Flavones	Human chondrocytes from OA patients with total knee replacement surgery, OA model in mice	Alleviated cartilage damage and proteoglycan loss, decreased MMP-13 and increased collagen-II expression in OA mice [61]
Chrysin	Flavonoids	Flavonols	In vitro OA IL-1β treated chondrocytes	Inhibited NO production, MMP-1, -3 and -13 expression, but also suppressed ADAMTS-4, -5 and blocked NF-kB activation [62]
Baicalin	Flavonoids	Flavones	Human OA IL-1β treated chondrocytes, OA model in mice	Inhibited COX-2, iNOS, MMP-3, -13 expression and NO, PGE2 production; relieved synovitis in OA mice [63]
Wogonin	Flavonoids	Flavones	Human OA IL-1β treated chondrocytes, cartilage explants	Switched the catabolic action to elevated expression of anabolic factors, suppressed oxidative stress and inflammation [64]
Apigenin	Flavonoids	Flavones	Differentiated THP-1 cells activated with sodium urate	Inhibited IL-1 β production and apoptosis-associated speck-like protein oligomerization, and improved the inflammatory symptoms associated with inflammasome activation [31]
Kaempferol		Flavonols		
3′, 4′-Dichloroflavone		Flavones		
Curcumin	Flavonoids	Curcuminoids	Orally taken capsules	Improved WOMAC and decreased inflammation in OA patients [28]
			In vitro IL-1β-stimulated human chondrocytes	Anti-apoptotic effect in chondrocytes and regulation of cartilage degradation through MMP-3, caspase-3 and IL-1 β actions [65]
			In vitro incubation in human osteoarthritic chondrocytes	Suppressed the oxidative stress-induced responses involved in OA pathogenesis [35]
Oxymatrine	Alkaloid	Quinolizidine alkaloids	In vitro incubation in human synoviocytes	Decreased IL-6 and IL-8 expression through of NF-kB inhibition [66]
Curcumin and resveratrol	Polyphenols combination	Curcuminoids and phytoalexins	In vitro incubation in human articular IL-1 β treated chondrocytes	Synergistically inhibited catabolic effect, MAPK pathway and apoptosis of chondrocytes [67]

**Table 2 pharmaceutics-13-01108-t002:** Lipid nanostructures encapsulating natural compounds with anti-inflammatory activity.

Natural Compound	Lipid Nanostructure	Composition	Administration	Activity [Reference]
*Polysaccharides*				
GLU	Liposomes	DSPC:GLU 8:2 (molar ratio)	In vitro studies in primary mouse chondrocytes	Accelerated cell viability and proliferation; down-regulation of pro-inflammatory cytokines; up-regulation of anabolic components [151]
CS	Positive Liposomes	PC:DOPE:Chol:SA 4:2:3:1 (molar ratio) Size—170.3 nm; PDI—0.218; ζ—10.44 mV	In vitro studies in stressed L929 mouse fibroblasts	Protective effect against oxidative damage and decrease of pro-inflammatory cytokines production [152]
CS	Positive liposomes embedded in type I collagen and freeze-dried	PC:DOPE:Chol:SA 4:2:3:1 (molar ratio) E.E.—68.2% Size—523.83 nm; PDI—0.40; ζ—10.44 mV	L929 mouse fibroblasts injected in sterile freeze-dried matrices	Better control of CS release compared to liposomal CS; allowed cell penetration for regenerative activity [153]
CS and GLU	Liposomes	Epikuron 200©, Epikuron 200© SH	Oral administration to rabbits	High permeation through intestinal mucosa; no histopathological alterations of the intestinal tissue [154]
CS and tapentadol	CS surface modified nanovesicles	PC:Chol:SA 7:3:1.5 (molar ratio)	Sublingual administration in OA-induced Wistar rats	Improved bioavailability and reduction of pain [155]
CS and diacerein	CS modified SLN	Lecithin:SA 1:6.25 (mass ratio)	IA administration in femoro-tibial joint of rat knee	Increased drug bioavailability [156]
HA	Liposomes	PC: HA solution (1:1, volume ratio)	Surface force balance measurements	The model suggested that multiple lipid layers formed on the surface increased lubrication, while HA could be complexed by lipids in the synovial fluid [125]
HA and celecoxib	Liposomes embedded in HA gel	PC:Chol 5:1 (mass ratio)	Single IA administration in rabbit OA knee model	Effective in pain control and cartilage protection [157]
*Proteins*				
Leech saliva extract rich in proteins and peptides	Liposomes	PC:Chol 95:5 (mass ratio)	Topical administration in human OA patients for 1 month	Enhanced skin absorption; 50% pain relief; reduction of joint inflammation and stiffness [158]
Lactoferrin	Positive MVL (Multivesicular liposomes) Liposomes	DPPE:Chol:SA 5:5:1 (molar ratio)	IA administration in CIA DBA1 mice	Prolonged the residence time for better reduction of inflammation, compared to free lactofferin; decreased pro-inflammatory cytokines production (TNF-α, IFN-γ); increased anti-inflammatory cytokines (IL-5, IL-10) [23,150]
*Polyphenols*				
Quercetin and resveratrol	Liposomes	Lipoid S75:oleic acid 10:1 (mass ratio)	Oxidative stressed fibroblasts	High cellular uptake and superior ROS scavenging, compared to free polyphenols [159]
Resveratrol and curcumin	Lipid-core nanocapsules	Polycaprolactone:seed oil:sorbitan stearate 1:1.65:0.385 (mass ratio)	In vitro model of human primary chondrocytes treated with nitric oxide-donor to mimic OA joint conditions	Higher dose delivery, protective effect on cell morphology and membrane surface [30]
Morin	Mannose decorated Liposomes	DSPC:Chol: F-DHPE 60:35:5 (molar ratio)	Arthritic rats treated intravenously for 3 days	Preferential internalization into macrophages, inhibited the osteoclastogenesis, pro-inflammatory cytokines (TNF-α, IL-1β, IL-6, IL-17), VEGF angiogenic factor and iNOS inflammatory enzyme production; suppressed RANKL and STAT-3 expression, but increased osteoprotegerin expression [160]
p-Coumaric acid	Mannose decorated Liposomes	DSPC:Chol:mannose 60:35:5 (molar ratio)	Ex vivo studies in macrophages; arthritic rats treated intravenously, for 3 days	Targeted synovial macrophages, inhibited osteoclasts differentiation, suppressed expression of MMP-9 and inflammatory cytokines [161]
Oxymatrine	Positive MV liposomes	EPC:Chol:DSPE-PEG 4:1:1 (mass ratio) E.E.—73.4%; Size—178 nm; PDI—0.167; ζ—13.30 mV	Intraperitoneal administration in intervertebral disc degeneration (IVDD) mice model	Reduced the mRNA and protein level of MMP3, MMP-9 and IL-6 [92]
Curcumin	Liposomes	PC:Chol 30:70 (molar ratio)	In vitro studies in 7F2 osteoblasts and RAW 234.7 macrophages	High cellular uptake, favored osteoblast differentiation and mineralization, increased OPG:RANKL ratio and prevented osteoclastogenesis [162]
Curcumin	Liposomes	Lecithin:Chol 18:1 (molar ratio)	Sistemic administration in C57BL/6J mice with hemi-lung radiation	Inhibited NF-kB pathway, down-regulated pro-inflammatory cytokines TNF-α, IL-6 and IL-8, and TGF-β [163]
Curcumin	Liposomes	DMPC:DMPG:Chol 7:1:8 (molar ratio)	In vitro studies of liposomal curcumin in human blood, plasma and culture medium of human lymphocytes, splenocytes and virus-transformed human B-cells	Higher stability and inhibitory effects on concanavalin A-stimulated human lymphocytes, splenocytes and B-cells proliferation; better bioavailability and efficacy, compared to free curcumin, recommending its clinical application [164]
Curcumin (curcuminoids)	Exosomes	Exosomes: curcumin 1:2:9 (mass ratio)	In vitro studies in RAW 264.7 macrophages and in vivo studies in C57BL/6J mice with oral administered or injected system	Increased bioavailability and anti-inflammatory activity of curcumin [165]
Dimethyl curcumin	Liposomes	PC:Chol 4:4 (molar ratio)	IA injections (six times) in CIA rats model	Regulated gelatinases release and cell cycle of spleen lymphocytes [166]

## Data Availability

Not applicable.

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
