# Peer review of "Mechanisms and Pharmaceutical Action of Lipid Nanoformulation of Natural Bioactive Compounds as Efficient Delivery Systems in the Therapy of Osteoarthritis"

_pharmaceutics, 2021, doi:10.3390/pharmaceutics13081108_

Round 1
Reviewer 1 Report
pharmaceutics-1257146
Review
Liposomes as Efficient System for Intra-articular Delivery of Natural Bioactive Compounds in the Therapy of Osteoarthritis
The aim of this review was to present the design and characteristics of the liposomes as lipid nanostructured carriers and controlled release systems of natural bioactive compounds with pharmaceutical properties and to discuss their efficacy in the treatment of OA using intra-articular (IA) delivery.
This review focus on liposomes as lipid nanostructured delivery systems for some natural bioactive compounds of animal and plant origin that have been proposed or examined to date for the treatment of OA.
The study was structured as follows:
Natural Bioactive Compounds with Anti-inflammatory Activity and Mechanisms of Action:
Polysaccharides with Anti-inflammatory Activity, Plant Polyphenols with Anti-inflammatory Activity
Recent Advances in Liposomes Technology: Novel Methodological Approaches, Advantages and Limits of using Liposomes as Efficient Delivery Systems, IA vs. Oral Administration
Liposomes Loaded with Natural Anti-inflammatory Compounds for OA Treatment
Anti-inflammatory Activity and Mechanisms of Action in in Vitro and in Vivo Experimental Models was studied: Redox Control, Modulation of Catabolic Mediators, Modulation of Cytokines Level, Modulation of Inflammasome and Toll-like Receptors (TLR), Control of Bone Remodelling Processes
As an observation, the text is very difficult to follow due to the abbreviations
Author Response
Reviewer 1:
Thank you very much for reviewing this manuscript and also for your comments.
As an observation, the text is very difficult to follow due to the abbreviations.
An Abbreviations section is provided in the revised manuscript at page 19, line 593, before References.
Reviewer 2 Report
This manuscript presents the review on liposomes to deliver bioactives originated from nature such as flavonoids, curcuminoids, etc. It specifically covers the intra-articular delivery for osteoarthritis. It covers the recent literature with justified numbers of articles reviewed.
In general, the manuscript is nicely organized and written. I recommend this for publication after minor corrections.
Title: author must ensure that intraarticular delivery is used this term only if manuscript specifically discuss intraarticular delivery. I feel this term in title might be misleading as it may be correlated with intraarticular injection. However, the manuscript covers in vitro and in vivo studies from different routes, where some are intraarticular injections. Please change and improve the title to avoid misleading.
Tables: I suggest including abbreviations below the tables that are used in table. This will help the readers.
The information in table related to liposomes is minimal. It can include the composition of liposomes, and their size characteristics of the liposomes such as size, PDI, zeta.
Discussion of literature on how different types of natural bio actives are loaded based on their solubility, stability, etc, will be useful.
I suggest including some relevant figures for better understanding and to attract wide readership.
Author Response
Thank you for your valuable comments that improved the manuscript.
Title: author must ensure that intraarticular delivery is used this term only if manuscript specifically discuss intraarticular delivery. I feel this term in title might be misleading as it may be correlated with intraarticular injection. However, the manuscript covers in vitro and in vivo studies from different routes, where some are intraarticular injections. Please change and improve the title to avoid misleading.
The title of the manuscript was modified to “Mechanisms and Pharmaceutical Action of Liposomal Formulation of Natural Bioactive Compounds in the Therapy of Osteoarthritis”, in order to better describe the content.
Tables: I suggest including abbreviations below the tables that are used in table. This will help the readers.
An Abbreviations section is provided in the revised manuscript at page 19, line 593, before References.
The information in table related to liposomes is minimal. It can include the composition of liposomes, and their size characteristics of the liposomes such as size, PDI, zeta.
Table 2 at page 11 was completed with information about liposomes composition and other liposomes’ characteristics (lipid molar ratio, type, size, homogeneity, surface charge) provided by the authors.
Discussion of literature on how different types of natural bioactives are loaded based on their solubility, stability, etc, will be useful.
I suggest including some relevant figures for better understanding and to attract wide readership.
Discussion on hydrophilic and lipophilic incorporation into liposomes was initially given at page 2 (lines 81-82), page 9 (line 280-282), page 9 (line 303-305). At Section 4 (page 10), we have added the phrase “In general, natural bioactive compounds with anti-inflammatory properties lack water solubility and chemical stability, but liposomal formulations of their derivatives showed improved therapeutic benefits [141].” and a new reference [141] Coimbra, M.; Isacchi, B.; van Bloois, L.; Torano, J.S.; Ket, A.; Wu, X.; Broere, F.; Metselaar, J.M.; Rijcken, C.J.F.; Storm, G.; Bilia, R.; Schiffelers, R.M. Improving solubility and chemical stability of natural compounds for medicinal use by incorporation into liposomes. Int J Pharm 2011, 416, 433-442.
Reviewer 3 Report
Major concern:
According to the title “Liposomes as Efficient System for Intra-articular Delivery of Natural Bioactive Compounds in the Therapy of Osteoarthritis” and this review aims to present the design and characteristics of the liposomes carrier systems for natural bioactive compounds. However, the authors focus mainly on the mechanism of these natural bioactive compounds and their pharmaceutical action in the whole manuscript.
Only some general liposome technology and usage are described in sections 3 and 4 and summarized in Table 2. It is not the same as the title and purpose of this article.
It is necessary to strengthen the advantages and disadvantages of different liposome formulation designs as delivery systems for natural biologically active compounds.
Author Response
Thank you for reviewing the manuscript and for your valuable comments.
According to the title “Liposomes as Efficient System for Intra-articular Delivery of Natural Bioactive Compounds in the Therapy of Osteoarthritis” and this review aims to present the design and characteristics of the liposomes carrier systems for natural bioactive compounds. However, the authors focus mainly on the mechanism of these natural bioactive compounds and their pharmaceutical action in the whole manuscript.
Only some general liposome technology and usage are described in sections 3 and 4 and summarized in Table 2. It is not the same as the title and purpose of this article.
The title of the manuscript was modified to “Mechanisms and Pharmaceutical Action of Liposomal Formulation of Natural Bioactive Compounds in the Therapy of Osteoarthritis”, in order to better describe the content. Also, the Abstract was revised for a better correlation with the aim of this review.
It is necessary to strengthen the advantages and disadvantages of different liposome formulation designs as delivery systems for natural biologically active compounds.
Advantages and limits of liposomal formulation were provided and discussed as a separate Section 3.2., at page 9. In addition, the advantages were highlighted at page 2 (lines 83-85), page 11 (lines 383-385), page 18 (lines 577-578).
Reviewer 4 Report
The work represents an interesting overview of the use of liposomes as drug delivery.
There is a recent and comprehensive literature search.
I recommend publishing
Author Response
Thank you for your comments.
Round 2
Reviewer 3 Report
- Since the title “Liposomes as Efficient System for Intra-articular Delivery of Natural Bioactive Compounds in the Therapy of Osteoarthritis” have been changed to "Mechanisms and Pharmaceutical Action of Liposomal of Natural Bioactive Compounds in the Therapy of Osteoarthritis", it looks more fit to the main focus on the mechanism of natural bioactive compounds and their pharmaceutical action.
- There still have some spelling mistakes in the whole manuscript that need to be corrected.
Author Response
Thank you for reviewing the manuscript and for your valuable comments.
- Since the title “Liposomes as Efficient System for Intra-articular Delivery of Natural Bioactive Compounds in the Therapy of Osteoarthritis” have been changed to "Mechanisms and Pharmaceutical Action of Liposomal of Natural Bioactive Compounds in the Therapy of Osteoarthritis", it looks more fit to the main focus on the mechanism of natural bioactive compounds and their pharmaceutical action.
Thank you for your comments.
- There still have some spelling mistakes in the whole manuscript that need to be corrected.
We have revised again the spelling according to your suggestions.
